# Interplay of Extracellular Vesicles and TLR4 Signaling in Hepatocellular Carcinoma Pathophysiology and Therapeutics

**DOI:** 10.3390/pharmaceutics15102460

**Published:** 2023-10-13

**Authors:** Stavros P. Papadakos, Konstantinos Arvanitakis, Ioanna E. Stergiou, Christos Vallilas, Stavros Sougioultzis, Georgios Germanidis, Stamatios Theocharis

**Affiliations:** 1First Department of Pathology, School of Medicine, National and Kapodistrian University of Athens, 10679 Athens, Greece; stavrospapadakos@gmail.com; 2First Department of Internal Medicine, AHEPA University Hospital, Aristotle University of Thessaloniki, 54636 Thessaloniki, Greece; arvanitak@auth.gr; 3Basic and Translational Research Unit (BTRU), Special Unit for Biomedical Research and Education (BRESU), Faculty of Health Sciences, School of Medicine, Aristotle University of Thessaloniki, 54636 Thessaloniki, Greece; 4Department of Pathophysiology, School of Medicine, National and Kapodistrian University of Athens, 10679 Athens, Greece; stergiouioa@med.uoa.gr (I.E.S.); ssougiou@med.uoa.gr (S.S.); 5Molecular Oncology Unit, Department of Biological Chemistry, Medical School, National and Kapodistrian University of Athens, 10679 Athens, Greece; chris-vallilas@hotmail.com

**Keywords:** PAMPs, DAMPs, TLR4, exosomes, extracellular vesicles, HCC, immunotherapy

## Abstract

Hepatocellular carcinoma (HCC) stands as a significant contributor to global cancer-related mortality. Chronic inflammation, often arising from diverse sources such as viral hepatitis, alcohol misuse, nonalcoholic fatty liver disease (NAFLD), and nonalcoholic steatohepatitis (NASH), profoundly influences HCC development. Within this context, the interplay of extracellular vesicles (EVs) gains prominence. EVs, encompassing exosomes and microvesicles, mediate cell-to-cell communication and cargo transfer, impacting various biological processes, including inflammation and cancer progression. Toll-like receptor 4 (TLR4), a key sentinel of the innate immune system, recognizes both pathogen-associated molecular patterns (PAMPs) and damage-associated molecular patterns (DAMPs), thereby triggering diverse signaling cascades and pro-inflammatory cytokine release. The intricate involvement of the TLR4 signaling pathway in chronic liver disease and HCC pathogenesis is discussed in this study. Moreover, we delve into the therapeutic potential of modulating the TLR4 pathway using EVs as novel therapeutic agents for HCC. This review underscores the multifaceted role of EVs in the context of HCC and proposes innovative avenues for targeted interventions against this formidable disease.

## 1. Introduction

### 1.1. Hepatocellular Carcinoma Epidemiology and Management

Hepatocellular carcinoma (HCC) is the most common type of primary liver cancer and a major cause of cancer-related mortality worldwide [1]. According to GLOBOCAN, HCC is the sixth most common cancer worldwide and the third leading cause of cancer-related deaths. It is estimated that approximately 840,000 new cases of liver cancer were diagnosed globally in 2020, and HCC accounted for approximately 75% to 85% of all primary liver cancers [2]. The incidence of HCC varied significantly by geographic region and underlying risk factors. HCC is more common in men than women, with a male-to-female ratio of approximately 2.8:1 [3]. The prognosis for HCC is generally poor, with a 5-year survival rate of approximately 18%. The mortality rate is also high. with estimated 800,000 deaths per year worldwide [2]. Nevertheless, early identification and intervention can enhance the results and augment the chances of survival. Prevention measures for HCC include reducing exposure to risk factors such as hepatitis B (HBV) and C viruses (HCV), avoiding alcohol and tobacco use, and implementing vaccination programs for hepatitis B [3]. The National Comprehensive Cancer Network’s (NCCN) guidelines for managing HCC adapted to different resource settings suggest screening individuals at high risk using both liver ultrasonography and α-fetoprotein (AFP) [4].

### 1.2. Extracellular Vesicles: Biogenesis and Functions

Extracellular vesicles (EVs) constitute vesicles originating from cell membranes and encompassing cargo such as proteins, nucleic acids, lipids, metabolites, and even organelles from parental cells [5,6,7]. The categorization of EVs includes exosomes, microvesicles, and apoptotic bodies, as stipulated by the International Society of Extracellular Vesicles [8]. These entities lack a self-replication ability and emerge through distinct mechanisms of biogenesis [9]. Exosomes, with dimensions ranging from 30 to 120 nm, are formed intracellularly within multivesicular bodies (MVBs). The release of exosomes into the extracellular space occurs through the fusion of MVBs with the plasma membrane [10]. Microvesicles, spanning from 50 to 1000 nm, are generated when cells directly bud vesicles from the plasma membrane [9]. Apoptotic bodies, measuring 50 to 5000 nm, emerge from the cell membrane and are primarily released by cells undergoing apoptosis [9]. The main differences in cargo composition among these various extracellular vesicles are related to their biogenesis and size [11]. Exosomes are typically smaller and enriched in specific proteins and various RNA species, making them important mediators of intercellular communication. Microvesicles, while also carrying proteins and nucleic acids, are larger and have a broader range of cargo. Apoptotic bodies, on the other hand, mainly contain cellular debris from dying cells. These differences in cargo composition reflect the distinct roles these vesicles play in cell communication and physiology [11]. The notions discussed earlier are depicted in Figure 1.

EVs fulfill a wide spectrum of biological functions, participating in various physiological and pathological processes [12,13]. When facilitating intercellular communication by transferring a diverse array of molecules between cells, EVs play pivotal roles in intricate biological events like tumorigenesis [6], the formation of pre-metastatic niches [14], inflammation, and immune modulation [15]. Their composition reflecting the state of the parent cell during production renders them promising candidates for diagnostics [16,17]. As they remain stable in numerous biological fluids and are abundant, EVs offer significant potential as biomarker reservoirs [7]. Circulating EVs within liquid biopsies could enable prognosis monitoring, disease progression tracking, and therapy response assessment [7,18]. Recent state-of-the-art reviews emphasize the significant contribution of EVs produced by HCC cells to the progression and spread of HCC [7]. Glycolytic enzymes like Rab20 and triosephosphate isomerase 1 (TPI1), as well as enzymes like caspase-3 and neutral sphingomyelinase 1 (NSMase1), have been shown to have a significant impact on HCC growth and invasion [19,20]. p120ctn and Lysyl Oxidase Like 4 (LOXL4)-containing EVs influence cell migration and proliferation [21]. Additionally, the 14-3-3ζ protein in EVs impairs tumor-infiltrating T lymphocyte functions, while complement factor H (CFH) and circular RNAs (circRNAs) regulate immune evasion and cell signaling [22,23].

### 1.3. The Toll-like Receptor 4 Signaling Pathway

Toll-like receptor 4 (TLR4) is a transmembrane protein that plays a crucial role in innate immune response by recognizing and responding primarily to lipopolysaccharides (LPSs) from bacterial cell walls [24]. LPSs are complex molecules found in the outer membrane of Gram-negative bacteria, which are composed of three main regions: lipid A, core oligosaccharide, and O antigen [25]. The lipid A moiety is a biologically active component of LPS. In addition to LPS, TLR4 can also respond to other molecules derived from both pathogens and host cells [26]. This includes lipoteichoic acid (LTA), which is a component of the cell wall of Gram-positive bacteria that activates TLR2 and TLR4 signaling, viral envelope proteins such as the respiratory syncytial virus (RSV) fusion protein and the HCV envelope protein alongside endogenous ligands released during tissue damage or inflammation such as heat shock proteins, the high mobility group box 1 (HMGB1) protein, and extracellular matrix components such as hyaluronan and synthetic ligands including monophosphoryl lipid A (MPLA) and glucopyranosyl lipid A (GLA) [26].

#### 1.3.1. Ligand Recognition and TLR4 Activation

LPS recognition by TLR4 requires the TLR4 co-receptor myeloid differentiation factor 2 (MD2) [27]. The extracellular LPS binding protein (LBP) interacts with the bacterial outer membrane, leading to an alteration that facilitates the extraction of single LPS molecules via CD14, which, in turn, transfers a single LPS molecule to the MD2 protein. Once LPS is transferred to MD2 via CD14, TLR4 dimerization takes place. A functional LPS receptor comprises TLR4-MD2 heterodimers. This complex then recruits adapter proteins, initiating TLR4 downstream signaling via two main pathways [27].

The MyD88-Dependent Pathway:

The MyD88-dependent pathway is the first and most well-known pathway activated by TLR4 [28]. Upon recruitment to the TLR4-MD2 complex, MyD88 interacts with the cytoplasmic domain of TLR4, leading to the recruitment and activation of interleukin (IL)-1 receptor-associated kinase 1 (IRAK1) and IRAK4. This results in the phosphorylation and activation of tumor necrosis factor receptor-associated factor 6 (TRAF6), which, in turn, activates the downstream transcription factor nuclear factor-kappa B (NF-κB). NF-κB then translocates to the nucleus and induces the expression of pro-inflammatory cytokines, such as tumor necrosis factor (TNF)-α, IL-1β, and IL-6.

The MyD88-Independent Pathway:

The MyD88-independent pathway is also known as the toll-interleukin-1 receptor (TIR)-domain-containing adaptor protein that induces the interferon-β (TRIF)-dependent pathway as it requires TRIF for downstream signaling [28]. Upon ligand binding, the TLR4-MD2 complex recruits TRIF, which activates a series of kinases, including TANK-binding kinase 1 (TBK1) and the inhibitor of nuclear factor kappa-B kinase ε (IKKε). These kinases then phosphorylate and activate the transcription factor interferon regulatory factor 3 (IRF3), which translocates to the nucleus and induces the expression of type I interferons (IFN-α and IFN-β). Figure 2 provides a concise illustration of the aforementioned content.

#### 1.3.2. Negative Regulators of TLR4 Signaling

To prevent the excessive or prolonged activation of TLR4 signaling, several negative regulators have been identified [28,29]. These include the single immunoglobulin (Ig) IL-1 receptor-related molecule (SIGIRR), which acts as a decoy receptor and inhibits TLR4 signaling, the suppressor of cytokine signaling 1 (SOCS1) which negatively regulates the MyD88-dependent pathway, A20 which is a protein that can inhibit the activation of TLR4 signaling by removing ubiquitin chains from signaling molecules such as TRAF6, the toll-interacting protein (TOLLIP) which binds to the TLR4 receptor and prevents the recruitment of downstream signaling molecules and IRAK-M which is an inhibitor of IRAK1 and IRAK4 [29]. In addition, several microRNAs, such as microRNA-146b, have been identified to target TLR4 signaling components and modulate the magnitude and duration of the response [30].

In the context of HCC, mounting evidence has emerged, pointing toward an intricate interplay between EVs and TLR4 pathways [31]. TLR4 signaling is a multifaceted player in HCC, influencing metastasis, drug resistance, epigenetic regulation, proliferation, apoptosis, and angiogenesis [31]. TLR4 signaling in HCC immunotherapy involves enhancing the effectiveness of treatments like atezolizumab and bevacizumab or tremelimumab and durvalumab. It has shown promise in reshaping the tumor microenvironment and promoting anti-tumor immune responses [31]. TLR4 signaling can improve HCC cancer vaccines, regulate the immune landscape of the HCC microenvironment, and impact various immune cells, including T cells, B cells, dendritic cells, neutrophils, myeloid-derived suppressor cells, and macrophages [31]. Targeting TLR4 signaling may enhance the efficacy of the PD-1 blockade and other immunotherapies in HCC treatment [31]. Utilizing extracellular vesicles as a platform for drug delivery is a burgeoning area of research and development [18]. The field of EVs has seen rapid evolution, spanning from their discovery in 1967 to their current applications in diagnostics, therapeutics, and drug delivery systems. Future prospects for EVs in drug delivery are promising, with standardized procedures and new insights into donor cell types and drug-loading techniques expected to drive clinical successes [32]. The use of EVs in HCC treatment and the delivery of nucleic acids and small molecule drugs are on the rise [32]. For example, hepatocyte-derived EVs enriched with saturated fatty acids activate TLR4 signaling, promoting pro-inflammatory responses and hepatocyte insulin resistance [33,34]. In a similar way, the exploitation of EVs for the delivery of various molecules that could interfere with TLR4 signaling has increased interest in the field of HCC. Our review comprehensively summarizes the basic aspects of TLR4 signaling and targeting via EVs in HCC and explores how this can be therapeutically exploited.

## 2. The Molecular Mechanisms of LPS Tolerance

The liver in humans receives 1.5 L of blood every minute, originating from two sources: the portal vein and the hepatic artery. This blood supply carries an enormous antigenic load, consisting of harmless dietary and commensal products that the hepatic immune system must tolerate [35]. Simultaneously, the immune system in the liver must respond to a variety of blood-borne pathogens, such as viruses, bacteria, and parasites, as well as metastatic cells that frequently target the liver [36]. To address this challenge, the liver requires tight immune regulation, or the so-called LPS/endotoxin tolerance [37]. In essence, the liver has a complex and unique system that enables it to manage the dual challenge of tolerating harmless antigens and responding appropriately to pathogenic challenges [36,38]. Several cellular mechanisms, such as immunosuppressive cells and molecules, including cytokines and ligands, are present in abundance in the liver to ensure that pathogen products and antigens typically do not stimulate immune responses.

To begin with, numerous non-parenchymal cells (NPCs) including liver sinusoidal endothelial cells (LSECs), Kupffer cells (KCs) and dendritic cells (DCs) have the ability to present antigens to T cells in ways that promote exhaustion [39,40], leading to the expression of inhibitory receptors on T cells such as programmed cell death protein 1 (PD-1) and T-cell Ig mucin domain-containing protein 3 (Tim-3) [41]. Secondly, another key mechanism of immune tolerance in the liver involves liver DCs. The liver contains multiple subsets of DCs, including myeloid DCs (mDCs) and plasmacytoid DCs (pDCs). These cells are specialized for antigen presentation and can induce either a state of tolerance or effective immunity [42]. In the liver, mDCs express programmed cell death ligand-1 (PD-L1), which drives the activation and expansion of classic FoxP3^+^ CD25^+^ CD4^+^ T regulatory (Treg) cells [43], which have been shown to suppress liver allograft rejection. DCs can also promote Treg development through their expression of indoleamine-pyrrole 2,3-dioxygenase (IDO), an enzyme that catabolizes tryptophan and generates an immunosuppressive product (kynurenine) [44]. Another important mechanism of immune tolerance in the liver is the role of KCs, which are resident macrophages of hepatic sinusoids. KCs can endocytose fragments of damaged hepatocytes, which convey transforming growth factor (TGF)-β1 and cause KCs to secrete IL-10 [45]. IL-10 acts in an autocrine manner, leading KCs to induce a variety of immunosuppressive mechanisms, including both effector T cell suppression and Treg cell promotion [40,46]. KCs also express B7-H1 (PD-L1), a ligand that engages the PD-1 receptor on activated T cells, resulting in clonal exhaustion [47]. Continuous exposure to LPS from the intestine may cause the down-regulation of TLR4-signaling pathways in KCs, resulting in LPS tolerance. This tolerance is partly induced by iIRAK-M [48]. Regarding LSECs, they respond to LPS initially by expressing TLR4/CD14, but with repeated stimulation, they become unresponsive to LPS while still retaining their scavenger activity. The LPS tolerance observed in LSECs is marked by a decrease in the nuclear translocation of NF-κB upon subsequent LPS exposure, which is intricately associated with the production of prostanoids. LPS tolerance in LSECs results in reduced leukocyte adhesion following LPS rechallenge and improved sinusoidal microcirculation in the liver, contributing to the local hepatic control of inflammation [49]. The molecular mechanisms that have been associated with the suppression of TLR4 signaling after prolonged LPS stimulation [50] involve changes in gene expression, signaling pathways, and epigenetic modifications and are analyzed in depth elsewhere [37]. These mechanisms include reduced LPS-induced MyD88-TLR4 association [51], reduced IRAK activity [51], enhanced nuclear factor of kappa light polypeptide gene enhancer in B-cells inhibitor α (IkBα) degradation [52] and reduced activation of p38, extracellular signal-regulated kinase (ERK), c-Jun N-terminal kinase (JNK), and/or NF-κB [53]. Additionally, negative feedback regulators of TLR4 signaling, such as A20 [54,55] and IRAK-M [56], have been found to delay the onset of LPS tolerance and could contribute to the suppression of gene expression during tolerance [37]. Moreover, NF-κB subunits and other transcription factors that are activated by LPS are also altered during tolerance. In naive cells, LPS stimulation leads to the loss of histone H3K9 demethylation [57] and an increase in H4 acetylation, allowing for SWItch/sucrose non-fermentable (SWI/SNF) complex recruitment and gene transcription. However, in tolerized cells, a Trichostatin A (TSA)-sensitive histone deacetylase may prevent H4 acetylation. Sirt1 facilitates the recruitment of RelB to promoters of tolerizable genes [58,59], subsequently engaging G9a to inhibit the expression of tolerizable genes by countering H3K9 demethylation [60]. For non-tolerizable gene promoters, G9a demethylates H3K9 in naive cells through the basal association of Atf7 [61]. Subsequent to tolerization, Atf7, and G9a are eliminated from these promoters, leading to the loss of H3K9 demethylation, which permits gene expression. Furthermore, latent enhancers might allow for the rapid expression of some genes during tolerance [62]. It is important to note that these changes have been directly shown to occur for only particular subsets of tolerizable and non-tolerizable genes and specific gene subsets, which likely differ in the specifics of their regulation, although the general paradigms of histone modification and transcription factor recruitment may be broadly applicable [37]. TLR4-signaling pathway alterations during LPS tolerance are summarized in Figure 3.

## 3. The Effects of TLR4 Signaling in Hepatocarcinogenesis

In recent years, there has been increasing interest in the role of TLR4 signaling in the development of HCC. Studies have suggested that TLR4 signaling might be involved in several aspects of hepatocarcinogenesis, including tumor growth, invasion, and metastasis. Yu et al., in a landmark study, investigated the relationship between endotoxin accumulation and liver tumorigenesis in rodents. They found that reducing endotoxin accumulation can prevent liver tumorigenesis in animal models. Specifically, they used antibiotics in rats or the genetic ablation of TLR4 in mice to reduce LPS levels, which prevented excessive tumor growth and multiplicity. They also revealed that TLR4 ablation sensitized the liver to carcinogen-induced toxicity via blocking NF-κB activation and sensitizing the liver to reactive oxygen species (ROS)-induced toxicity but lessened inflammation-mediated compensatory proliferation. Restoring myeloid cells expressing TLR4 in mice lacking TLR4 reinstated the hepatic inflammation and proliferation triggered by diethylnitrosamine (DEN), underscoring the paracrine mechanism of tumor promotion facilitated by LPS. Overall, this study provides valuable insights into the role of endotoxin accumulation in liver tumorigenesis and suggests potential avenues for cancer prevention [63].

### 3.1. The TLR4 Signaling in HCC Senescence

Senescence is a cellular process during which cells cease to divide and undergo irreversible growth arrest, constituting a hallmark of aging, while autophagy is a cellular process during which cells recycle and remove damaged or dysfunctional components to maintain cellular homeostasis [64]. Research has shown that autophagy plays a critical role in regulating senescence, as defective autophagy can lead to the accumulation of damaged cellular components that contribute to cellular aging and senescence [65]. Additionally, autophagy has been shown to regulate the senescence-associated secretory phenotype (SASP), which is a pro-inflammatory response that contributes to age-related diseases [66].

Wang et al., demonstrated that TLR4 activity regulates the expression of the DNA repair protein Ku70 which contributes to preventing the development and progression of HCC [67]. The study was conducted using a DEN-induced HCC mouse model with wild-type and *Tlr4* mutant mice. The findings suggest that TLR4-controlled immunity supports senescence induction and the expression of DNA repair proteins, which play an integrated defense role against genotoxic carcinogenesis and tumor progression in the liver. The research also examined whether inhibiting TLR4 activity through genetic or pharmacological means can lead to immune suppression and thereby restrict the development and advancement of tumors in the liver. The results show that the inhibition of TLR4 activity leads to immune suppression, which limits tumorigenesis and tumor progression in the liver. These findings suggest that TLR4 activity induces programmed cell death, maintains intracellular senescent responses to avoid excessive proliferation and malignant transformation, maintains an effective autophagy flux to clear toxic p62-positive aggregates, interrupts its feedback with accumulated ROS, and enhances the expression of DNA repair proteins, such as Ku70, to eliminate the risk of genome instability. Therefore, the authors highlight several implications for the prevention and treatment of HCC [67]. Wang et al., further documented that repairing DNA damage by X-Ray Repair Cross Complementing 6 (XRCC6)/KU70 could restore senescence and autophagic flux by reducing DNA damage and restoring both the TP53-cyclin-dependent kinase inhibitor (CDKN1A)/p21- and CDKN2A/p16-RB1/pRb-dependent cellular senescence [68]. This prevented the runaway replication of damaged hepatocytes. Additionally, repairing DNA damage caused a broad-spectrum increase in immune responses which activated autophagy as indicated by the elevated expression of microtubule-associated protein 1A/1B-light chain 3 (LC3)-I/-II, Beklin 1 (BECN1), phosphatidylinositol-4,5-bisphosphate 3-kinase catalytic subunit 3 (PIK3C3) and degradation of sequestosome 1 (SQSTM1). These changes were all associated with the enhanced activity of the TLR4-mediated-p38 mitogen-activated protein kinase (MAPK)/NF-κB and IRF3 signaling. Hence, the TLR4-mediated immune response plays a role in promoting both senescence and autophagy [68].

### 3.2. TLR4 Signaling in Innate Cellular Populations

Neutrophils are often found in high numbers near human tumors and in mouse models of cancer, but their role in cancer is unclear as they seem to promote both tumor growth and clearance [69,70]. This could be explained by the existence of different subtypes of neutrophils or the diversity of the tumor microenvironment (TME). Neutrophil extracellular trap (NET)-induced coagulation is now associated with cancer and promotes a tumorigenic microenvironment [71]. NETs can have both positive and negative effects on cancer, inducing clotting in tumor blood vessels that may be beneficial for tumor destruction or accelerating the metastatic process by trapping tumor cells and allowing them to move through vessels [70]. Inhibiting the action of DNase could result in the disintegration of NETs, leading to a decrease in the degree of metastasis. Zhan et al., conducted a clinical study aiming to investigate whether NETs are involved in HBV-related hepatocarcinogenesis and have clinical significance in the evaluation and management of HCC. The study included 175 individuals diagnosed with HCC, both with and without HBV infection, as well as 58 healthy controls [72]. They found that HCC patients, especially those infected with HBV, had elevated levels of NETs in the blood serum and tissue specimens. NETs facilitated HCC growth and metastasis by promoting angiogenesis, epithelial–mesenchymal transition (EMT)-related cell migration, matrix metalloproteinase (MMPs)-induced extracellular matrix (ECM) degradation, and NETs-mediated cell trapping. Blocking the generation of NETs through DNase-1 was found to effectively prevent HCC growth and the metastasis caused by them. The generation of NETs was accelerated by HBV-induced S100A9 and mediated by the activation of TLR4/RAGE-ROS signaling. In HBV-related HCC, the levels of circulatory NETs were found to be correlated with viral load, TNM stage, and metastasis status. These findings suggest that NETs could serve as a potential biomarker to predict extrahepatic metastasis in HBV-related HCC [72].

In the liver, macrophages are present in two different forms: KCs, which are tissue-resident macrophages, and infiltrating macrophages, which migrate from the blood to the tumor site [73]. Tumor-associated macrophages (TAMs) are the most abundant population. TAMs are evolutionarily linked to micro-vessel density in tumor tissues and can be classified as M1-like or M2-like based on their anti-tumor or pro-tumor activity in the TME, respectively. Studies have revealed that an abundance of M2-like TAMs in the TME can lead to tumor immunosuppression and chemoresistance, while a higher M1-like to M2-like TAM ratio is associated with better long-term outcomes in cancer patients [74]. TLR4 signaling in macrophages is believed to play a significant role in the pathogenesis of HCC by promoting tumor growth, ΕΜΤ and inflammation. Hepatic macrophages are the primary cells responsible for the production of “cancer-promoting cytokines,” such as IL-6 and TNF-α, in response to LPS [73]. Miura et al., documented in a NASH-related HCC model that the levels of these cytokines are higher in hepatic macrophages from Pten^Δhep^ mice than in those from Pten^fl/fl^ mice, while the former were more susceptible to a low concentration of LPS than peritoneal macrophages in Pten^Δhep^ mice. These findings demonstrate the high susceptibility of hepatic macrophages in Pten^Δhep^ mice to LPS. The expression of these cytokines was reduced in tumor-suppressed groups, including Pten^Δhep/Tlr4−/−^ mice and Tlr4^−/−^ BM-transplanted chimeric mice, due to the absence of TLR4 signaling in macrophages [75]. IL-6 and TNF-α induce the proliferation of oval cells, which are putative cancer progenitor cells that appear in severe liver injury [76]. Furthermore, Miura et al., demonstrated that IL-6 promoted the proliferation of oval cells and Huh 7 cells. In vivo, non-tumor tissue showed the emergence of epithelial cells expressing the stem cell marker epithelial cell adhesion molecule (EpCAM), which increased in number under the conditions of sustained inflammation in Pten^Δhep^ mice. Cancer progenitor cells were observed in Pten^Δhep^ mice before tumor development, and isolated cancer progenitor cells could promote tumor formation. Therefore, the pro-inflammatory cytokines produced by macrophages promote tumor growth by inducing the proliferation of cancer progenitor cells and tumor cells in Pten^Δhep^ mice [75]. Thus, targeting TLR4 appears to be a promising approach for preventing HCC. Yao et al., analyzed the crosstalk between HCC and non-tumor cells in the TME with an emphasis on the role of M2-polarized macrophages [77]. In particular, Yao et al., suggested that M2-polarized macrophages upregulate TLR4 expression in HCC cells and activate the signal transducer and activator of the transcription 3 (STAT3) signaling pathway downstream of TLR4 which could be one of the primary mechanisms for metastasis promotion. In more depth, the upregulation of TLR4 intensified the cancerous characteristics of SMMC-7721 and MHCC97-H cells that were grown in a conditioned medium from M2-like macrophages (M2-CM) while the utilization of TLR4-neutralizing antibodies resulted in a notable inhibition of M2-CM-induced EMT, and migration in transwell migration assays [77]. TLR4 promotes cancer progression by activating several signaling pathways [78], but in this particular study, M2 macrophages specifically activated the STAT3 signaling pathway [77]. STAT3 is correlated with invasiveness and poor prognosis [79]. TLR4/STAT3 formed a critical axis that was successively activated by M2-polarized macrophages in HCC cells [77], comprising an ideal target as an anti-metastatic therapeutic agent. Taking a step further, the cornerstone of specific and individualized immunotherapy is to reveal the expression profile of a specific immune checkpoint on immune cells in the TME. Studies have shown that patients with a high expression of PD-1 in CD8^+^ T cells in the tumor-immune microenvironment can achieve better clinical efficacy when treated with PD-1 monoclonal antibodies [80]. Similarly, the glucocorticoid-induced tumor necrosis factor receptor (GITR) expression profile in the HCC microenvironment needs to be elucidated for DTA-1 (a GITR-agonistic antibody) therapy [81]. Pan et al., found that different patients presented different types of GITR expression profiles, which could explain why different individuals respond differently to DTA-1 treatment. Additionally, this study demonstrated that GITR is mainly highly expressed in tumor-infiltrating Tregs, while low expression is found in CD8^+^ T cells, macrophages, naive T cells, etc. Tumor-infiltrating (Ti)-Tregs were considered a preferable potential target for DTA-1 treatment [81]. The authors also documented how macrophages play an important role in DTA-1 drug resistance and increased M2 polarization becomes the main mechanism of DTA-1 resistance. Furthermore, they showed how the expression profile of GITR was mainly concentrated in T cells rather than macrophages. Instead, M2 polarization mainly depended on T cells rather than Treg cells, and the GITR-ligand could directly increase Th2 differentiation and reduce Foxp3 expression. Th2-mediated M2 polarization and Ti-Treg reduction were the main biological reactions caused by DTA-1 therapy, with the former being the main mechanism of DTA-1 resistance, and the latter serving as a potential force for antitumor effects. To overcome DTA-1 resistance in HCC, M2 polarization needs to be targeted, and TLR4 agonists are proposed to reverse M2 polarization. Combining TLR4 agonists with DTA-1 treatment provides a promising therapeutic strategy to solve DTA-1 resistance [81].

To conclude, TLR4 signaling in innate immune cellular populations in HCC TME comprises a potential biomarker and a candidate therapeutic target to prevent HCC, tackle the metastatic process, and reverse drug resistance in DTA-1 treatment.

Figure 4 summarizes the effects of TLR4 signaling in hepatocarcinogenesis both in the context of hepatocytes and tumor-infiltrating immune cells.

## 4. The Role of TLR4 Signaling in HCC Patients

Over the past few years, there has been a significant increase of interest in understanding the potential role of TLR4 in the pathogenesis and progression of HCC. Although the precise mechanisms by which TLR4 is involved in HCC development and progression are still being actively investigated, research findings indicate that this receptor may play a crucial role in the pathophysiology of HCC. In fact, some studies have identified TLR4 as a promising clinical biomarker for HCC.

### 4.1. TLR4 Polymorphisms in HCC

Jiang et al., examined the association between a functional variant (rs1057317) at the microRNA-34a (miR-34a) binding site in the tlr4 gene and the risk of HCC [82]. They conducted a case–control study at a single center, genotyping *TLR4* sequence variants using the polymerase chain reaction (PCR) and direct sequencing in 426 hepatocellular carcinoma cases and 438 controls. A luciferase activity assay was used to measure the impact of rs1057317 on the binding of hsa-miR-34a to the TLR4 messenger RNA (mRNA). The study found that individuals with the AA genotype for rs1057317 were significantly more likely to develop HCC than those with the wild-type homozygous CC genotype (adjusted odds ratio [OR] ranging from 1.116 to 2.452, *p* = 0.013). The reporter vector activity was lower in the vector carrying the C allele than in the one carrying the A allele. The expression of TLR4 was also detected in the peripheral blood mononucleated cells of HCC patients, indicating that mRNA and protein levels of TLR4 could be influenced by the single nucleotide polymorphism (SNP) rs1057317. These findings suggest that the risk of developing HCC may be linked to a functional variant at the miR-34a binding site in the tlr4 gene and that the miR-34a/TLR4 axis may play a crucial role in hepatocellular carcinoma development [82]. Zahran et al., determined the prognostic significance of TLR2 and TLR4 expression on circulating monocytes in patients with HCC [83]. They enrolled 40 Egyptian patients with radiologically diagnosed hepatic focal lesions as HCC and 38 age and sex-matched healthy controls. They showed that the expression of both TLR2 and TLR4 on monocytes was significantly higher in HCC patients than in the controls. There was a positive correlation observed between the levels of alanine transaminase (ALT), aspartate transaminase (AST), and AFP, TLR2, and TLR4 expression. Conversely, there was an inverse correlation observed between TLR2 and TLR4 expression and overall survival (OS). Moreover, the high expression of TLR2 was significantly associated with poor response to treatment, while the high expression of both TLR2 and TLR4 was linked to poor OS. These findings suggest that the increased expression of TLR2 and TLR4 on peripheral monocytes could indicate the development and progression of HCC and can serve as a prognostic marker [83]. Further research on the role of TLRs in HCC pathogenesis and prognosis is needed to identify potential therapeutic targets. Androutsakos et al., in a single-center cohort study, investigated the association of TLR4 SNPs (a transition at SNP rs4986790 resulting in an Asp/Gly polymorphism at amino acid 299 and a transition at SNP rs4986791 resulting in a Thr/Ile polymorphism at amino acid 399) with HCC occurrence and all-cause and liver-related mortality, as well as the time between the diagnosis of cirrhosis and HCC development or the diagnosis of HCC and death [84]. Of the 260 patients enrolled, 52 had or developed HCC. TLR4 SNPs did not show any correlation with the primary or secondary endpoints, except for a shorter duration between HCC development and death in patients with TLR4 mutations. Overall, TLR4 SNPs did not demonstrate any correlation with carcinogenesis or death in patients with liver cirrhosis. Patients with TLR4 SNPs who developed HCC had lower survival rates: a finding that requires further investigation [84]. Finally, Shi et al., investigated the relationship between the TLR4 rs1927914 polymorphism and HCC recurrence following liver transplantation (LT). The aim of this study was to assess whether the TLR4 gene rs1927914 polymorphism of donors and recipients is associated with HCC recurrence after LT. A total of 83 patients with HCC who underwent LT from July 2006 to June 2015 were included. The researchers genotyped an SNP (rs1927914) in both donors and recipients and evaluated the association between the polymorphism and the risk of tumor recurrence. They showed that the donor TLR4 rs1927914 polymorphism was significantly associated with HCC recurrence after LT. They confirmed that Milan criteria, microvascular invasion, and the donor TLR4 rs1927914 genotype were independent risk factors for HCC recurrence. Patients carrying donors homozygous for TT had significantly lower recurrence-free survival (RFS) and OS than CC/CT patients [85].

A growing body of clinical studies has shown that TLR4 expression and activation are increased in patients with viral hepatitis-induced HCC [86,87] and that elevated serum levels of soluble TLR4 may be a useful biomarker for HCC detection and monitoring [88]. Further studies are needed to better understand the mechanisms of TLR4 involvement in HCC development and progression and to explore potential therapeutic targets for HCC treatment. A brief overview of data on viral hepatitis-related HCC is presented in Table 1.

### 4.2. TLR4 as Biomarker in HCC

TLR4 has emerged as a potential biomarker in HCC. Several studies have investigated the role of TLR4 in HCC and its potential as a diagnostic and prognostic marker.

In a clinical study which involved 30 patients who had been diagnosed with HCC at the University Hospital Central of Asturias in Oviedo, Spain, Eiro et al., demonstrated that there was a positive association between the co-expression of TLR3, TLR4, and TLR9 in terms of tumor cells and tumor size. Patients with tumors that were both TLR4- and TLR9-positive in immunostaining (IHC) in HCC cells had a poor prognosis [94]. Kang et al., observed an upregulation of TLR4 expression in HCC tissues. The positive rate of TLR4 in HCC tissues was 77.8%, while adjacent noncancerous tissues only had a rate of 20%, indicating an upregulation of TLR4 expression in HCC. Furthermore, TLR4 expression was found to be correlated with sex and a lower TNM stage but not with age, tumor size, or the differentiation of patients. The study also found a positive correlation between TLR4 expression and the levels of IL-17A and IL-23, which are key mediators of inflammation that contribute to carcinogenesis, suggesting a possible regulation effect between the expression of TLR4 and that of IL-17A and IL-23 in HCC [95]. Wang et al., utilized IHC to examine the expression of TLR4 in HCC tissues. The normal liver exhibited weak staining for TLR4 in hepatocytes and cholangiocytes. Non-cancerous adjacent tissues from HCC patients displayed weak-to-moderate staining for TLR4, with only a small proportion of hepatocytes exhibiting high-intensity staining. By contrast, a large proportion of cancer cells in HCC tissues showed increased TLR4 expression. However, there was no evidence of obvious membrane or nuclear staining in HCC cells or hepatocytes, as all cases displayed a cytoplasmic staining pattern. Out of the 53 samples analyzed, 44 showed positive TLR4 staining, with no correlations found between TLR4 expression and various clinicopathological characteristics. Nevertheless, TLR4 expression was found to be correlated with Ki-67 expression (*p* = 0.024) [96]. Surgical resection following radiotherapy (RT) in patients with locally advanced HCC may be appropriate. Wu et al., conducted a clinical study with 20 HCC patients who underwent post-RT surgery to assess the impact of TLR4 signaling. OS and disease-free survival (DFS) outcomes were evaluated for each patient, and radiation-induced liver diseases (RILDs) were detected. The survival analysis indicated that patients expressing low levels of TLR4 or TNF-related apoptosis-inducing ligand (TRAIL) had a better OS compared to those with high expression levels of TLR4 (*p* = 0.003) or TRAIL (*p* = 0.007). Patients with low expression levels of vascular endothelial growth factor receptor 2 (VEGFR2) or TRAIL had significantly longer median DFS compared to those with high expression levels of VEGFR2 (*p* = 0.003) or TRAIL (*p* = 0.008). However, the expression levels of TLR4, VEGFR2, or TRAIL in peritumoral liver tissue did not show any significant differences in OS or DFS times. However, patients with a high expression of these factors in peritumoral liver tissue post-RT showed more severe RILDs (*p* < 0.05). Thus, the expression levels of TLR4 and its associated proteins in HCC tumors could serve as prognostic factors for patients undergoing post-RT surgery, while the inhibition of TLR4, VEGFR2, and TRAIL expression in HCC and non-tumor liver tissue could potentially reduce the severity of RILDs and improve survival outcomes in the future [97]. Finally, the inflammatory processes initiated by acute liver graft injury orchestrated tumor recurrence, which is the primary hurdle when expanding LT for patients with HCC. Liu et al., documented that the elevated levels of C-X-C motif chemokine ligand 10 (CXCL10) and TLR4 in small-for-size liver grafts were found to be linked to tumor recurrence in patients with HCC after LT. At 2 h after reperfusion, the levels of CXCL10 (*p* = 0.0191) and TLR4 (*p* = 0.0091) within the graft were significantly higher in patients who received small-for-size grafts compared to those who received large grafts. The intragraft expression of CXCL10 (*p* = 0.0068) and TLR4 (*p* = 0.0056) was also significantly increased in patients who experienced HCC recurrence, as opposed to those who did not, which is consistent with the findings of myeloid-derived suppressor cell (MDSC) analysis. Additionally, the intragraft expression of CXCL10 and TLR4 was strongly correlated (*p* = 0.0019). The co-localization of TLR4 and the MDSC marker CD33 further indicated that CXCL10 may act through TLR4 to mobilize MDSCs after transplantation [98]. The above indicates that TLR4 could serve as a biomarker for HCC recurrence after LT. Collectively, while the results of the above studies are promising, further clinical studies are needed to determine the practical implications of these findings and to establish their applicability in a clinical setting.

## 5. Leveraging EV Influence on TLR4 Pathways in HCC

In the context of HCC research, a captivating avenue of investigation has emerged, centering on the intricate interplay between EVs and TLR4 pathways [31]. As scientific knowledge advances, the systematic examination of the effects mediated by EVs on TLR4 pathways emerges as a compelling field, effectively bridging the gap between intricate molecular communication and pioneering therapeutic strategies.

### 5.1. The Therapeutic Immune Modulation of TLR4 Signaling by EVs

This extensive review highlights TLR4 signaling’s intricate role in shaping the HCC immune landscape [31]. Recent interest in EVs as potential therapeutic targets underscores their novel impact on TLR4 pathways, offering a promising avenue for immune modulation in HCC. The interplay between EVs and TLR4 signaling pathways in the context of HCC is summarized in Figure 5 and analyzed in the following section.

It is widely known that exosomes derived from hypoxic bone marrow (BM)-derived mesenchymal stromal cells (Hp-MSCs) enriched with miR-182-5p play a role in promoting liver regeneration by targeting TLR4 and orchestrating macrophage polarization toward an anti-inflammatory state through the FOXO1 pathway [99]. Zhou et al., investigated the impact of HCC-derived EVs containing a long noncoding RNA (lncRNA) known as PART1 on HCC progression [100]. They showed that PART1 and TLR4 were upregulated, while miR-372-3p was downregulated in HCC tissues and cells. PART1 overexpression in HCC cells led to increased proliferation, migration, invasion, and EMT. Mechanistically, PART1 binds to miR-372-3p, reducing its expression. MiR-372-3p negatively targets TLR4 in macrophages. Furthermore, this study showed that EVs containing PART1 contributed to the M2 polarization of macrophages and the development of HCC by affecting the miR-372-3p/TLR4 axis. This suggests that HCC cell-derived EVs containing PART1 can upregulate TLR4 by inhibiting miR-372-3p, ultimately promoting the M2 polarization of macrophages in the context of HCC. In conclusion, this study unveiled a mechanism by which tumor-derived EVs (TDEs) carrying PART1 exerted an oncogenic effect on HCC. These EVs influence macrophage polarization into the tumor-promoting M2 phenotype, highlighting the blockage of this pathway as a potential avenue for therapeutic intervention [100]. Taking a step further, HMGB1 released under hypoxic stress activates caspase-1 via TLR4 and RAGE signaling pathways in HCC, inducing an inflammatory response that promotes invasiveness and metastasis [101]. Li et al., demonstrated that LPS-sensing pathways, specifically TLR4 and caspase-11 signaling, regulate HMGB1 release [102]. They documented that during endotoxemia, caspase-11 and gasdermin D (GsdmD) play a role in translocating HMGB1 from the nucleus to the cytoplasm via the calcium-induced phosphorylation of calcium-calmodulin kinase (camkk)β. Cleaved GsdmD accumulation on the endoplasmic reticulum led to calcium leak and an increase in intracellular calcium. Notably, they showed that the exosome pathway is crucial for HMGB1 release from hepatocytes, a process dependent on TLR4 and independent of caspase-11 and GsdmD. They unveiled a novel mechanism where TLR4 signaling enhanced caspase-11 expression and exosome release, while caspase-11/GsdmD activation contributed to HMGB1 accumulation in the cytoplasm via calcium release and camkkβ activation [102]. Along the same lines, another study explored the cellular and molecular mechanisms behind immune modulation associated with HCC [103]. Ye et al., documented a previously unrecognized protumorigenic subset of T cell Ig and mucin domain protein 1 (TIM-1)^+^ regulatory B (Breg) cells and thoroughly investigated their function, induction mechanisms, and clinical significance within the HCC TME. These TIM-1^+^Breg cells exhibited distinct characteristics compared to conventional peripheral Breg cells. They revealed that TDEs triggered the transformation of B cells into TIM-1^+^Breg cells via the HMGB1-TLR2/4-MAPK pathway. Remarkably, the high infiltration of TIM-1^+^Breg cells correlated with advanced disease stage and poor prognosis in HCC patients. TLR2/4 and MAPK inhibitors led to a decrease in the frequency of TIM-1^+^ B cells and inhibited the impact of TDE-induced B cells on CD8^+^ T cells. Εxploring these pathways, this study provides a potential avenue for targeted therapies in HCC treatment, potentially disrupting the immunosuppressive mechanisms orchestrated by TIM-1^+^Breg cells and the HMGB1-TLR2/4-MAPK pathway to counteract HCC progression [103]. In addition, Bretz et al., showed that exosomal signaling is TLR-dependent, as the knockdown of TLR2 and TLR4 blocks NF-κB and STAT3 activation, triggering the release of cytokines from mouse BM-derived DCsand macrophages and suggesting that exosomes stimulate TLR-dependent signaling pathways in monocytic precursor cells and possibly also in other immune cells [104]. This process could provide aid in the induction of immunosuppressive mechanisms during the progression of HCC. Finally, in a study exploring the significance of the S100A family of immune markers in HCC, S100A10 emerged as particularly relevant to HCC, affecting HCC proliferation through the annexin A2 (AnxA2)/Akt/mTOR signaling pathway [105]. S100A10 has a role in promoting HCC cell proliferation, migration, and invasion, suggesting its potential as a therapeutic target for HCC [106]. AnxA2 plays a crucial role in regulating the inflammatory responses triggered by TLR4 via the TRAM-TRIF pathway, facilitating TLR4 internalization and subsequent endosomal translocation, activating anti-inflammatory signaling and releasing cytokines [107], while the heterotetrameric complex of AnxA2 with S100A10 activates macrophages through TLR4 [108]. Wang et al., identified that S100A10 is secreted by HCC cells into EVs, enhancing stemness and metastatic properties, activating epidermal growth factor receptor (EGFR), AKT and ERK signaling, and promoting EMT. S100A10 regulates the protein cargos in EVs, mediating the binding of specific proteins to EV membranes through their interaction with integrin αV. Blocking EV-S100A10 with a neutralizing antibody can mitigate these effects [109]. The synergistic enhancement of these effects could potentially be achieved through the therapeutic manipulation of TLR4; however, further research in this direction is warranted. In summary, the above can be regarded as strategies aimed at modifying the immune tumor microenvironment in the context of HCC.

Finally, the interplay between microRNA-34a (miR-34a) and TLR4, along with the potential role of EVs, reveals complex dynamics in HCC initiation and progression. A functional variant, rs1057317, situated at the miR-34a binding site within the TLR4 gene, has been associated with an increased risk of HCC [82]. This variant not only impacts HCC risk but also influences the binding of miR-34a to TLR4 mRNA [82]. A Phase 1 study involving MRX34, a liposomal miR-34a mimic, aimed to determine the recommended Phase 2 dose (RP2D) of MRX34 and assess its efficacy in patients with refractory solid tumors [110]. Although the study observed common adverse events, RP2D was established for HCC and non-HCC cancers, demonstrating the dose-dependent modulation of target gene expression [110]. Furthermore, miR-34a emerges as a potent tumor suppressor, influencing the behavior of cancer stem cells (CSCs) by targeting key stemness factors [111]. Collectively, these interconnected findings shed light on the potential for novel therapeutic approaches in HCC treatment.

### 5.2. Strategic Approaches for HCC Prevention

In light of the prevalent association of HCC with fibrosis and cirrhosis, alongside efforts to modify the TME, a noteworthy avenue involves employing preventive strategies [112,113]. One such strategy entails harnessing EVs to target TLR4 signaling. This modulation holds the promise of curbing the progression of HCC by simultaneously adjusting the gut microbiota, enhancing intestinal barrier integrity, and suppressing liver inflammation.

It is widely accepted that the absence of TLR4 in hepatocytes dampens the inflammatory response induced by a high-fat diet (HFD), not only locally but also systemically. Despite the relatively subdued inflammatory response initiated by hepatocytes, chronic low-grade inflammation prompted by HFD was remarkably subdued in TLR4-deficient mice, impacting the liver, adipose tissue, and circulation [114]. Garcia-Martinez et al., investigated the role of small EVs enriched with saturated fatty acids (SFAs) in mediating crosstalk between hepatocytes and macrophages, leading to liver inflammation and hepatocyte insulin resistance in the context of NAFLD. Hepatocyte-released lipotoxic small EVs (sEVs) activated pro-inflammatory responses in macrophages via TLR4 activation. These lipotoxic sEVs, carrying SFAs, triggered rapid liver inflammation upon injection, contributing to JNK phosphorylation, NF-κB nuclear translocation, pro-inflammatory cytokine expression (IL-6, IL-1b, TNF), and immune cell infiltration. TLR4 inhibition mitigated sEV-mediated liver inflammation and hepatocyte insulin resistance. They underscored the significance of hepatocyte-derived sEVs as carriers of SFAs, driving liver inflammation and insulin resistance through a hepatocyte–macrophage crosstalk, which has implications for NAFLD-related lipotoxicity and consequently hepatocarcinogenesis [34]. Additionally, hepatic stellate cells (HSCs) play a central role in liver fibrosis by depositing excess ECM. Under normal conditions, quiescent HSCs store lipids. However, in response to chronic liver injury, they lose lipids, proliferate, and generate matrix proteins. Geng et al., revealed a bidirectional communication between HSCs and KCs, indicating that HSCs, especially in early activation, induced a proinflammatory phenotype in KCs (M1 macrophages) via the release of EVs. This effect was linked to TLR4 activation [115]. These findings highlight the interplay between HSCs and KCs and its implications for liver inflammation and fibrosis progression. In alignment with the above, Ohara et al., utilized EVs derived from amnion-derived mesenchymal stem cells (AMSCs) to explore their therapeutic potential when treating liver inflammation and fibrosis [116]. In a rat model of NASH and liver fibrosis, AMSC-EVs were shown to significantly reduce KC activation, inflammatory cytokine expression (TNF-α, IL-1β, IL-6), and HSC activation. This ameliorative effect was also observed in vitro, where AMSC-EVs inhibited both KC and HSC activation, along with suppressing the LPS/TLR4 signaling pathway [116]. These findings suggest that AMSC-EVs could serve as a novel therapeutic approach for chronic liver disease by mitigating inflammation and fibrogenesis through the modulation of HSC and KC activation.

The gut–vascular barrier (GVB) dysfunction plays a significant role in the progression of NASH [117]. The disrupted GVB contributes to the development of NASH by allowing bacterial translocation from the gut into the bloodstream [118]. This translocation could lead to increased levels of plasma endotoxins and the elevated expression of hepatic Toll-like receptors (TLR4/TLR9), triggering inflammation, liver injury, and fibrosis. In essence, the compromised GVB appears to facilitate the communication between gut inflammation and hepatic injury, exacerbating the NASH pathogenesis [118]. Analogously, the intestinal epithelial barrier damage induces liver and jejunum inflammation involving LPS/TLR4/NF-κB signaling and NLRP3 inflammasome pathways [119]. Lamas-Paz et al., focused on the impact of acute alcohol injury on the gut–liver axis and its role in hepatic inflammation [120]. Acute alcohol exposure led to damage in the intestinal epithelium, including the decreased expression of protective proteins (mucin-2) and disruption of tight junctions (zonula occludens-1). The gut microbiota composition was also altered, with changes in bacterial populations (decreased α-diversity) and altered bacterial composition (increase in the lactobacillus phylum and a decrease in the lachnospiraceae family). This study also explored the role of EVs released by intestinal epithelial cells (IECs) in the gut–liver axis. In vitro experiments showed that EVs released by IECs and exposed to alcohol-directed changes in hepatocyte morphology and lipid accumulation led to the upregulation of *tlr4*. EVs isolated from the blood of alcohol-exposed mice suggested bidirectional communication between the gut and liver [120]. Overall, they revealed that acute alcohol injury disrupted the gut–liver axis, altering intestinal barrier function of gut microbiota and leading to hepatic inflammation. Fizanne et al., focused on understanding the role of EVs derived from feces (fEVs) and circulating EVs (cEVs) in NAFLD and NASH [121]. They documented that while both NAFLD and NASH patients’ EVs contain prokaryotic and eukaryotic components, only NASH-fEVs exerted detrimental effects. NASH-fEVs were found to increase intestinal permeability by reducing the expression of tight junction proteins. These changes were linked to the activity of non-muscular myosin light chain kinase (nmMLCK). NASH-fEVs also induced inflammation in endothelial cells through the LPS/TLR4 pathway. NASH-fEVs and NASH-cEVs were found to activate HSCs, leading to the expression of proteins associated with inflammation and fibrosis in the liver. This suggests that these EVs play a role in promoting liver injury and inflammation. Interestingly, they also explored the bacterial origins of fEVs and found differences between NAFLD and NASH patients. The composition of these bacterial components might contribute to the varying effects of EVs in different stages of liver disease [121]. Overall, these findings highlight the involvement of EVs in mediating barrier dysfunctions that contribute to liver injury, emphasizing the significance of nmMLCK and LPS carried by fEVs in these processes.

It is widely accepted that the disruption of GVD during diet-induced dysbiosis is a key factor leading to bacterial translocation and the development of NASH, with obeticholic acid emerging as a protective agent against barrier disruption and NASH progression [122]. In the same direction, Engevik et al., investigated the role of fusobacterium nucleatum, a bacterium found in the intestinal mucosa of patients with gastrointestinal disease, in promoting intestinal inflammation [123]. They demonstrated that F. nucleatum secreted outer membrane vesicles (OMVs) while containing compounds that activate proinflammatory cytokines in colonic epithelial cells. These effects were mediated by TLR4 and downstream effectors such as phospho (p)-ERK, p-cyclic AMP response element binding protein (CREB), and NF-κB. In mice with human microbiota, F. nucleatum induced inflammation, disrupted colonic architecture, and increased proinflammatory cytokine expression. Notably, intact microbiota protected against F. nucleatum-induced immune responses, revealing a mechanism by which F. nucleatum contributes to intestinal inflammation [123]. Conclusively, targeting OMVs and TLR4 could hold promise as therapeutic strategies to address the inflammatory effects initiated by Fusobacterium nucleatum. It is worth mentioning that the combined administration of mesenchymal stem cell-derived exosomes and glycyrrhetinic acid effectively mitigated acute liver ischemia-reperfusion injury by suppressing the TLR4 signaling pathway and inflammatory response [124].

Summarizing the above, the interplay between EVs and TLR4 pathways holds significant potential in HCC research. EVs’ modulation of TLR4 signaling emerges as a promising avenue for immune modulation in HCC, with studies showing the EV-mediated effects on TLR4 pathways that impact inflammation, macrophage polarization, and HCC progression. These studies reveal the intricate relationship between EVs and TLR4, offering insights into potential therapeutic strategies for addressing liver inflammation, fibrosis, and the prevention of HCC progression by targeting TLR4-related pathways.

## 6. Discussion

Immunotherapy has become the mainstay of treatment in HCC [125,126]. However, there are several challenges associated with immunotherapy for HCC [31,69,127,128]. One challenge is the immunosuppressive nature of the liver microenvironment, which can limit the effectiveness of immune checkpoint inhibitors and other immunotherapeutic agents [129]. A recently published clinical study found that the tumor immune barrier (TIB) structure, which involves SPP1^+^ macrophages and CAFs, is the primary structural feature related to immunotherapy resistance based on spatial transcriptomics (ST) and single-cell RNA-sequencing (scRNA-seq) data from patients receiving anti-PD-1 treatment [130]. SPP1^+^ macrophages tend to accumulate at the tumor boundary and participate in the formation of the TIB structure, which excludes T cells from infiltrating the tumor core. They also identified a group of tumor-specific fibroblasts associated with unfavorable outcomes in HCC patients, which promoted the formation of the ECM structure. Blocking SPP1 expression in mice with spontaneous liver cancer increased the infiltration of T lymphocytes into tumors, suggesting that targeting SPP1 could enhance the effect of anti-PD-1 immunotherapy [130]. Another challenge is the heterogeneity of HCC, which can lead to variable responses to treatment among patients [131,132,133]. In addition, the lack of validated biomarkers to predict a response to immunotherapy in HCC makes it difficult to identify which patients are most likely to benefit from treatment [134,135,136]. Finally, the high cost of immunotherapy and the limited availability of these treatments in certain regions of the world can also be significant challenges [137]. While immunotherapy has shown promise in the treatment of HCC, addressing these challenges is crucial for improving outcomes in patients with this disease. Toward this direction, TLR4 holds potential significance in the context of mRNA-based neoantigen vaccines for HCC immunotherapy [138]. Given that TLR4 is a key immune receptor, its activation by mRNA vaccines may enhance the immunogenicity of neoantigens. This activation can potentially bolster neoantigen-specific immune responses, contributing to effective antitumor immunity [138]. However, the complex balance between promoting immune responses and managing potential autoimmune toxicity in the context of HCC immunotherapy remains an essential consideration and warrants further exploration to optimize vaccine strategies.

Targeting TLR4 in HCC therapeutically may offer several potential benefits. TLR4 is a key receptor that plays a critical role in regulating the innate immune response to various stimuli, including microbial pathogens, danger signals, and endogenous ligands [24]. Notably, emerging research suggests that TLR4 also exerts a regulatory role in various cancers [139]. In HCC, TLR4 is overexpressed and has been associated with tumor growth, metastasis, and resistance to chemotherapy [96]. Thus, targeting TLR4 could have several potential therapeutic effects, such as the induction of tumor cell death. TLR4 activation on cancer cells can induce cell death through multiple mechanisms, including apoptosis and necroptosis [140]. Second, TLR4 activation on immune cells, such as dendritic cells and macrophages, can enhance anti-tumor immune response by promoting the activation and proliferation of T cells. Targeting TLR4 could, therefore, increase the efficacy of immunotherapy in HCC [77]. Third, TLR4 activation has been associated with resistance to chemotherapy in various cancers, including HCC. Targeting TLR4 could sensitize tumors to chemotherapy and improve treatment outcomes [141]. Overall, targeting TLR4 in HCC therapeutically has the potential to induce tumor cell death, enhance the anti-tumor immune response, and sensitize tumors to chemotherapy. Collectively, the integration of nanotheranostics holds immense promise in the realm of HCC diagnosis and treatment, with a strategic focus on targeting the TLR4 receptor [142].

The therapeutic potential of EVs in HCC is a subject of increasing scientific interest and holds promise for innovative diagnostic and therapeutic interventions [32,143,144]. Within the context of HCC, EVs have emerged as key players in the intricate network of intercellular communication within the tumor microenvironment [15,145,146]. These vesicles carry a cargo of bioactive molecules with the capacity to profoundly influence various aspects of HCC progression [109], encompassing tumor growth [147], immune evasion mechanisms [148,149], and resistance to therapeutic interventions. The distinctive composition of EV cargo, enriched with HCC-specific biomolecules, offers an attractive avenue for targeted therapeutic strategies [11]. Emerging approaches center on utilizing EVs as delivery vehicles for therapeutics, with the potential to directly engage HCC cells, impede tumor proliferation, hamper angiogenesis [150], and sensitize HCC cells to conventional treatment modalities [151]. As this field advances, unlocking the full therapeutic potential of EVs in HCC holds the promise of obtaining transformative strategies that can combat this challenging cancer.

However, more research is needed to fully understand the therapeutic potential of targeting TLR4 in HCC and to develop safe and effective TLR4-targeting therapies [31]. One major concern is the risk of compromising the immune system’s ability to fight infections. TLR4 is crucial for detecting and responding to bacterial infections, and blocking TLR4 signaling may increase susceptibility to bacterial infections. Additionally, TLR4 is expressed in many different cell types, and blocking its signaling may have unintended effects on other organ systems. Therefore, while TLR4 is an attractive target for HCC therapy, it is important to carefully consider the potential drawbacks of targeting this receptor. Finally, another potential disadvantage of targeting TLR4 as a therapeutic strategy for HCC is the lack of clinical studies investigating the safety and efficacy of TLR4 antagonists in HCC patients. While preclinical studies have shown promising results, including reduced tumor growth and improved survival, it is essential to conduct clinical trials to confirm these findings and evaluate the safety and effectiveness of TLR4 antagonists in humans. Without these studies, it is a challenge to determine the appropriate dose, timing, and duration of treatment required for optimal therapeutic benefits while minimizing potential adverse effects. Therefore, further research is necessary to determine the feasibility of targeting TLR4 as a therapeutic approach for HCC.

## 7. Conclusions

In conclusion, the TLR4 signaling pathway plays a critical role in liver physiology and pathology, including the development and progression of hepatocellular carcinoma. While preclinical studies have demonstrated the potential of targeting TLR4 as a therapeutic strategy in HCC, clinical studies are needed to validate its efficacy and safety. Exploring the therapeutic potential of EVs in HCC stands as a frontier with the potential to revolutionize the management of HCC in the near future. Therefore, further investigation into the TLR4 signaling pathway and its modulation may offer new avenues for the treatment and prevention of HCC: a devastating disease with limited therapeutic options.

## Figures and Tables

**Figure 1 pharmaceutics-15-02460-f001:**
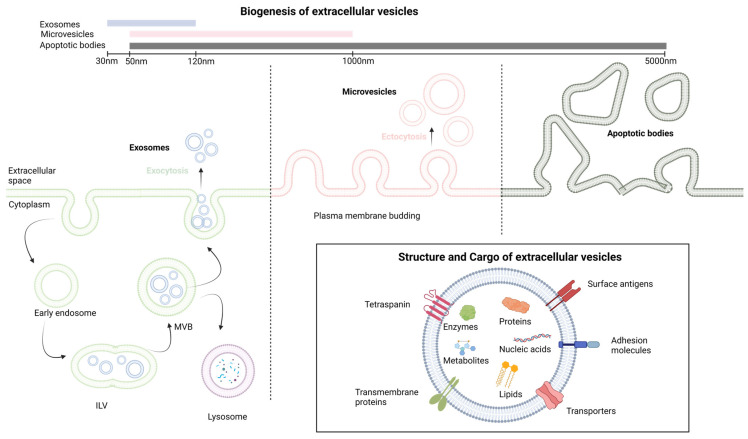
The fundamental aspects of extracellular vesicles structure and biogenesis. ILV, intralluminal vesicle; MVB, multivesicular body. Created with BioRender.com.

**Figure 2 pharmaceutics-15-02460-f002:**
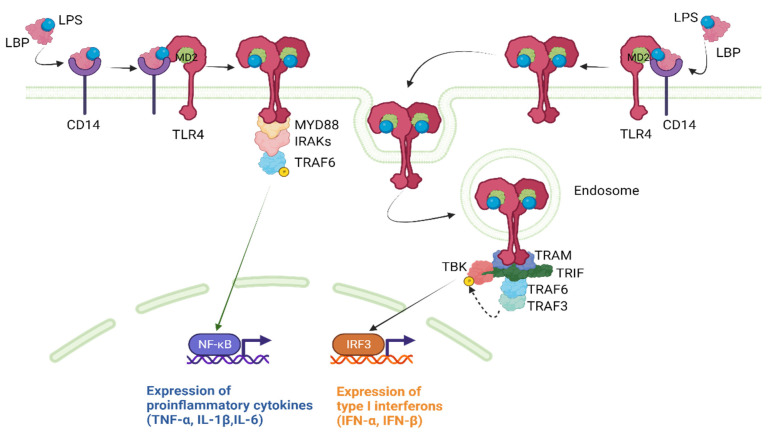
TLR4 downstream signaling pathways. IFN, interferon; IL, interleukin; IRAKs, IL-1 receptor-associated kinases; IRF3, interferon regulatory factor 3; LBP, lipopolysaccharide binding protein; LPS, lipopolysaccharide; MD2, myeloid differentiation factor 2; NF-κΒ, nuclear factor-kappa B; TBK1, TANK-binding kinase; TLR4, Toll-like receptor 4; TNF-α, tumor necrosis factor α; TRAF, tumor necrosis factor receptor-associated factor; TRAM, TRIF-related adaptor molecule; TRIF, TIR-domain-containing adaptor protein inducing interferon-β. Created with BioRender.com.

**Figure 3 pharmaceutics-15-02460-f003:**
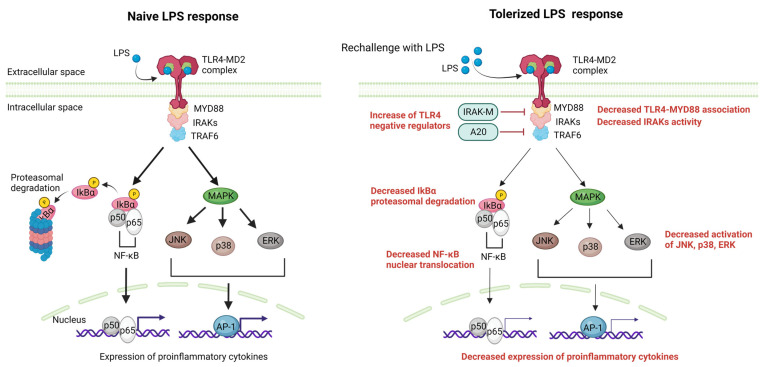
The liver manages a complex antigenic load from the blood, relying on immune tolerance mechanisms and mainly LPS tolerance. Myd88-dependent pathways activated after LPS engagement to the LPS-MD2 complex are negatively regulated in the case of LPS rechallenge. The main mechanisms include the increased activity of TLR4 negative regulators (i.e., IRAK-M,A20), decreased TLR4-Myd88 association, decreased IRAKs activity, decreased IkBα degradation resulting in decreased NF-κΒ nuclear translocation, and the decreased MAPK-mediated activation of JNK,p38 and ERK, resulting in decreased AP-1 induced transcription. The net result of these alterations is a reduction in the expression of proinflammatory cytokines after repeated exposure to LPS. AP-1, activator protein 1; ERKs, extracellular signal-regulated kinase; IRAKs, IL-1 receptor-associated kinases; JNK, c-Jun N-terminal kinase; LPS, lipopolysaccharide; MAPK,mitogen-activated protein kinase; MD2, myeloid differentiation factor 2; NF-κΒ, nuclear factor-kappa B; TLR4, Toll-like receptor 4; TRAF, tumor necrosis factor receptor-associated factor. Created with BioRender.com.

**Figure 4 pharmaceutics-15-02460-f004:**
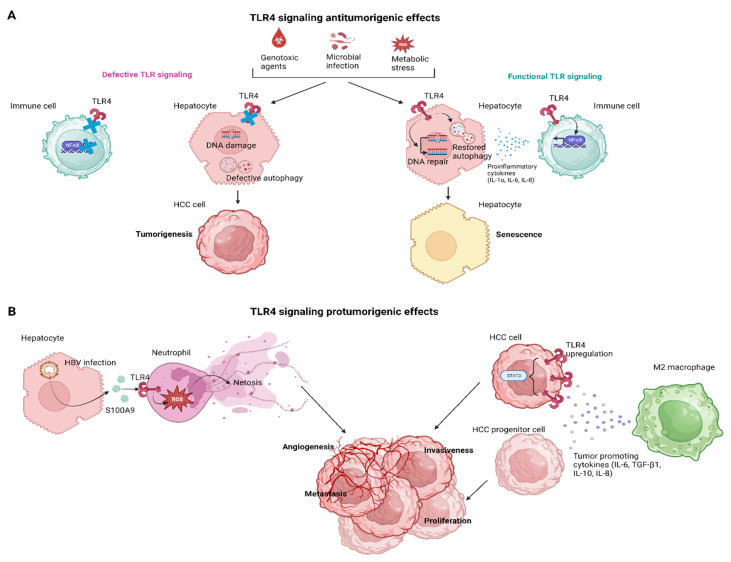
The effects of TLR4 signaling in hepatocarcinogenesis. TLR4 signaling in hepatocytes and immune cells can induce both antitumorigenic (**A**) and protumorigenic effects (**B**) during hepatocarcinogenesis. HBV, hepatitis B virus; HCC, hepatocellular cancer; IL, interleukin; NF-κB, nuclear factor-kappa B; ROS, reactive oxygen species; TGF-β, tumor growth factor; TLR, Toll-like receptor. Created with BioRender.com.

**Figure 5 pharmaceutics-15-02460-f005:**
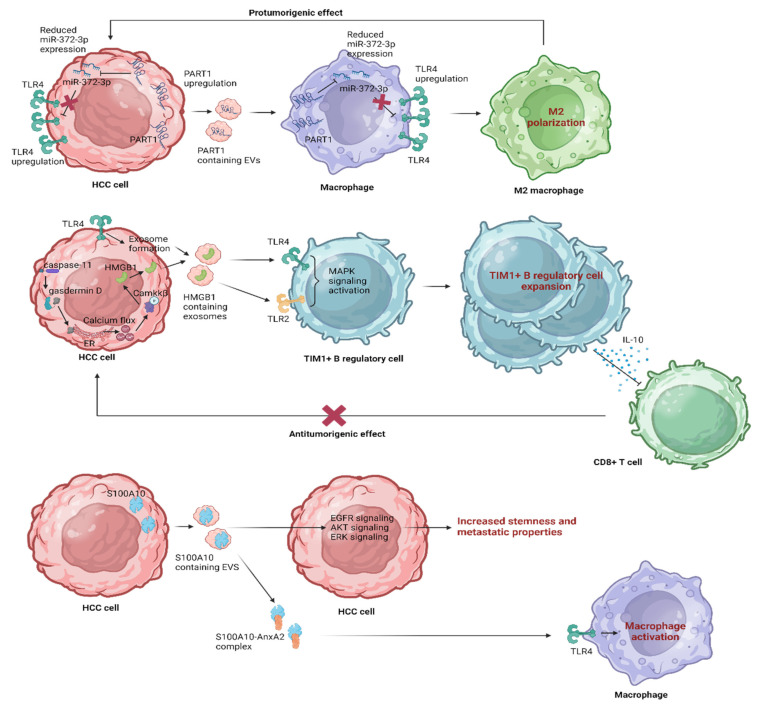
The interplay between EVs and the TLR4 signaling pathway in the context of HCC. EGFR, epidermal growth factor receptor; ER, endoplasmic reticulum; ERK, extracellular signal-regulated kinase; HCC, hepatocellular carcinoma; HMBGB1, high mobility group box 1; MAPK, mitogen-activated protein kinase; TLR, Toll-like receptor.

**Table 1 pharmaceutics-15-02460-t001:** Summary of data on the influence of TLR4 signaling in HBV/HCV-induced HCC.

Author/Year	Reference	Study Design	Outcomes
Agúndez et al. (2012)	[89]	rs2149356, rs4986791 and rs5030719 SNPs in 155 Patients with HCV-induced HCC, including 153 patients with HCV and 390 healthy controls	In comparison to the healthy controls, patients with HCC had a significantly lower frequency of the rs2149356 T allele carrier state (OR 0.421, 95% CI 0.285–0.625). This trend was also observed in patients with HCV (OR 0.426, 95% CI 0.236–0.767)
			The proportion of rs2149356 T allele carriers progressively decreased as the clinical stage of HCC increased
Al-Qahtani et al. (2014)	[90]	rs4986790 (A/G) and rs4986791 (C/T), in 450 HCV-infected and 600 uninfected controls	rs4986790 (A/G) and rs4986791 (C/T) had significantly different distributions between HCV-infected patients and uninfected controls (*p* < 0.0001; OR = 0.404, AC was found more frequently in chronic HCV-infected patients compared to cirrhosis/HCC patients (frequency = 94.7% and *p* = 0.04)
Neamatallah et al. (2019)	[86]	3295 individuals were divided into groups based on their HCV infection status and control subjects. Patients with liver cirrhosis and HCC	Haplotype CAGT of TLR4 was significantly associated with the CH and HCC groups
Sghaier et al. (2018)	[91]	174 HCV patients, 100 HBV patients and 360 healthy controls TLR4(rs4986790)	The minor (GG) genotype of TLR4 rs4985790 exhibited a notable positive correlation with HBV-associated HCC (*p* < 0.001)
Zhang et al. (2016)	[92]	949 HBsAg-positive patients: Group1-234 HBV carriers and CHB patients without cirrhosis or HCC; Group 2-281 cirrhotic patients without HCC and Group 3-434 cirrhotic patients with HCC	TLR4 SNP rs11536889 was found to be associated with an increased risk of HCC in patients with cirrhosis and CHB. TLR4 rs2149356 polymorphisms were linked to an increased risk of HCC in cirrhotic patients after conducting stratified analyses based on gender, age, and drinking history.
Salum et al. (2019)	[93]	493 blood samples for TLR4 rs4986791 SNP: 70 controls-252 SOF/DCV-treated HCV patients (65 HCC-187 patients did not develop HCC)-171 naıve HCV patients (48 early liver fibrosis, 21 late liver fibrosis and 102 HCC)	TLR4 rs4986791: the CC genotype was present in 100%, 81%, and 97% of EF, LF, and HCC patients, respectively. The minor protective TT genotype was completely absent in all subjects, while the CT genotype was absent in the EF group and was present in only 19% and 3% of LF and HCC patients, respectively (*p* = 0.001). The frequency of the protective T allele was higher in LF than in HCC (OR 0.047, 95% CI 0.005–0.43, *p* = 0.001)
Elkammah et al. (2020)	[88]	Soluble Toll-like receptor 4 (sTLR4) in 150 participants (50 HCV-related HCC, 50 HCV without HCC and 50 healthy controls)	sTLR4 levels in patients with HCV-related HCC (4436.1 ± 7089.8 pg/mL) compared to those with hepatitis C but without HCC (1561.4 ± 532.0 pg/mL) (*p* = 0.002) and the control group (1170.38 ± 159.42 pg/mL) (*p* < 0.001). Serum sTLR4 was positively correlated with serum alpha-fetoprotein levels and the tumor stages of HCC according to the Barcelona Clinic Liver Cancer staging system (BCLC). The combination of serum alpha-fetoprotein and serum sTLR4 increased the sensitivity of HCC detection to 76% and specificity to 94%.

## Data Availability

Not applicable.

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
