# Peer review of "Interplay of Extracellular Vesicles and TLR4 Signaling in Hepatocellular Carcinoma Pathophysiology and Therapeutics"

_pharmaceutics, 2023, doi:10.3390/pharmaceutics15102460_

Round 1

Reviewer 1 Report

The manuscript submitted by Stamatios Theocharis, Georgios Germanidis, and the co-authors is entitled "Exploring the Effectiveness of Extracellular Vesicles: Targeting TLR4 Signaling in Hepatocellular Carcinoma – Is There a Potential for Promising Therapeutic Benefit?"

Such unclear entitled reviews usually contain some problems, which, unfortunately, are also relevant to this article.

The authors describe in detail HCC and TLR4, but not the role of EVs in HCC.

The authors should explain what new information will get the reader from the article, perhaps the situation will become a little clearer

In its present form, the reviewer sees no other choice other than to reject the paper and recommend its resubmission with significant reduction of non-essential details, with a title corresponding to the contents, and formatted according to the requirements of the journal.

Sincelrey

Author Response

First Department of Pathology

Laiko General Hospital

National and Kapodistrian University of Athens

September 19th, 2023

Dear Editor,

Response: Exploring the Effectiveness of Extracellular Vesicles: Targeting TLR4 Signaling in Hepatocellular Carcinoma – Is There a Potential for Promising Therapeutic Benefit?

We thank you and the Reviewers for carefully evaluating our manuscript and for their positive and constructive feedback. Hopefully, our responses below address the points raised by the reviewers. All changes made are presented in the revised manuscript via the “track changes” option in MS word.

Reviewer 1: The manuscript submitted by Stamatios Theocharis, Georgios Germanidis, and the co-authors is entitled "Exploring the Effectiveness of Extracellular Vesicles: Targeting TLR4 Signaling in Hepatocellular Carcinoma – Is There a Potential for Promising Therapeutic Benefit?"

Such unclear entitled reviews usually contain some problems, which, unfortunately, are also relevant to this article. The authors describe in detail HCC and TLR4, but not the role of EVs in HCC. The authors should explain what new information will get the reader from the article, perhaps the situation will become a little clearer In its present form, the reviewer sees no other choice other than to reject the paper and recommend its resubmission with significant reduction of non-essential details, with a title corresponding to the contents, and formatted according to the requirements of the journal.

Response:

Dear Reviewer,

Thank you for your feedback and your valuable insights into our manuscript, "Exploring the Effectiveness of Extracellular Vesicles: Targeting TLR4 Signaling in Hepatocellular Carcinoma – Is There a Potential for Promising Therapeutic Benefit?"

We greatly appreciate your constructive comments, which have significantly contributed to improving the quality and focus of our manuscript. In response to your suggestions, we have restructured the article to provide a more coherent narrative that highlights the role of EVs in HCC. We have also clarified the new insights our article offers to readers, ensuring that the content aligns with the title.

Yours sincerely,

Stamatios Theocharis MD, PhD

Professor of Pathology

First Department of Pathology

Laiko General Hospital

National and Kapodistrian University of Athens

Mikras Asias 75, 11527, Athens, Greece

Tel: +30 2107462002

Email: [email protected], [email protected]

Reviewer 2 Report

This manuscript reviewed the interplay between EVs and TLR4 pathways in HCC research. The review is well written, however, there are some issues to consider

l   Line 138. “FoxP3+ CD25+ CD4+ Treg cells” should beFoxP3+ CD25+ CD4+ Treg cells”

l   Line 264. “CD8+ T cells” should be “CD8+ T cell”.

l   Line 537. “SPP1+ macrophages” should be “SPP1+ macrophages”

l   The title of this review is “Exploring the Effectiveness of Extracellular Vesicles: Targeting TLR4 Signaling in Hepatocellular Carcinoma – Is There a Potential for Promising Therapeutic Benefit?” I would suggest the authors to describe more about the therapeutic potential of EVs in HCC in the discussion and conclusion.

Author Response

First Department of Pathology

Laiko General Hospital

National and Kapodistrian University of Athens

September 19th, 2023

Dear Editor,

Response: Exploring the Effectiveness of Extracellular Vesicles: Targeting TLR4 Signaling in Hepatocellular Carcinoma – Is There a Potential for Promising Therapeutic Benefit?

We thank you and the Reviewers for carefully evaluating our manuscript and for their positive and constructive feedback. Hopefully, our responses below address the points raised by the reviewers. All changes made are presented in the revised manuscript via the “track changes” option in MS word.

Reviewer 2: This manuscript reviewed the interplay between EVs and TLR4 pathways in HCC research. The review is well written, however, there are some issues to consider

Response:

Dear Reviewer,

We would like to express our sincere gratitude for your kind words regarding our article. It is truly heartening to receive such positive feedback. Thank you for taking the time to review our article and for your constructive feedback.

Point 1.

Line 138. “FoxP3+ CD25+ CD4+ Treg cells” should be“FoxP3+ CD25+ CD4+ Treg cells”

Line 264. “CD8+ T cells” should be “CD8+ T cell”.

Line 537. “SPP1+ macrophages” should be “SPP1+ macrophages”

Response:  Your comments have been addressed

Point 2. The title of this review is “Exploring the Effectiveness of Extracellular Vesicles: Targeting TLR4 Signaling in Hepatocellular Carcinoma – Is There a Potential for Promising Therapeutic Benefit?” I would suggest the authors to describe more about the therapeutic potential of EVs in HCC in the discussion and conclusion.

Response: Thank you for your valuable feedback. We appreciate your constructive input, and we have taken your suggestions into consideration. In response to your comment, we have enriched the discussion and conclusion sections of our review to delve more comprehensively into the therapeutic potential of EVs in HCC. We believe that this enhancement strengthens the overall context of our review and underscores the significant promise that EV-based strategies hold potential therapeutic benefits in HCC. Your input is greatly appreciated.

Yours sincerely,

Stamatios Theocharis MD, PhD

Professor of Pathology

First Department of Pathology

Laiko General Hospital

National and Kapodistrian University of Athens

Mikras Asias 75, 11527, Athens, Greece

Tel: +30 2107462002

Email: [email protected], [email protected]

Reviewer 3 Report

In this manuscript, the authors summarized the progress of therapeutic potential of extracellular vesicles (EV) via TLR4 in HCC. The manuscript is well written, starting from briefly reviewing of EV biogenesis and types, followed by TLR4 pathways and its clinical significance in HCC, and finally with the introduction of EV effect on TLR4 signaling. Readers should be able to gain some overall knowledge in this topic.  Some minor issues the reviewer has for the authors to consider in the revision.

1. In the figure 1 and EV biogenesis and function part, the authors introduced three subtypes of EV, showing differences in biogenesis and size. It would be better to include some information whether there is any difference in contents between them.

2. In the figure 2, the authors show CD14 as a co-receptor of TLR4, whereas in the line 86, the authors use MD-2. The authors may want to keep consistent, at least, using aka such as MD-2 (CD14)?

3. In the LPS tolerance mechanism part, it would be better to have a figure to show the mechanisms  as described in the text part.

4. In the therapeutic potential of EV by targeting TLR4 in HCC or its prevention, the authors may have a short paragraph in introducing the utility of miR-34a in HCC treatment, given TLR4 is a target of TLR4. There are some previous studies showing the effect of miR-34a on tumor proliferation, invasion and metastasis (e.g., 10.1038/s41416-020-0802-1; PMID: 31114344; 10.3389/fcell.2021.640587).

5. line 554-556, the authors may make some comments on how TLR4 affect mRNA-based neoantigen vaccines in immunotherapy against HCC. 

Author Response

First Department of Pathology

Laiko General Hospital

National and Kapodistrian University of Athens

September 19th, 2023

Dear Editor,

Response: Exploring the Effectiveness of Extracellular Vesicles: Targeting TLR4 Signaling in Hepatocellular Carcinoma – Is There a Potential for Promising Therapeutic Benefit?

We thank you and the Reviewers for carefully evaluating our manuscript and for their positive and constructive feedback. Hopefully, our responses below address the points raised by the reviewers. All changes made are presented in the revised manuscript via the “track changes” option in MS word.

Reviewer 3: In this manuscript, the authors summarized the progress of therapeutic potential of EV via TLR4 in HCC. The manuscript is well written, starting from briefly reviewing of  EV biogenesis and types, followed by TLR4 pathways and its clinical significance in HCC, and finally with the introduction of EV effect on TLR4 signaling. Readers should be able to gain some overall knowledge in this topic.  Some minor issues the reviewer has for the authors to consider in the revision.

Response:

Dear Reviewer,

Thank you for taking the time to review our article and for providing us with your insightful feedback. We appreciate your positive comments regarding the overview of the current literature on EVs - TLR4 signaling in liver disease.

Point 1. In the figure 1 and EV biogenesis and function part, the authors introduced three subtypes of EV, showing differences in biogenesis and size. It would be better to include some information whether there is any difference in contents between them.

Response: Regarding your first point about Figure 1 and the EV biogenesis and function section, we appreciate your suggestion to include information about the differences in contents between the three subtypes of EV (exosomes, microvesicles, and apoptotic bodies). This is indeed an important aspect. Therefore, a relevant part has been included in the manuscript. Regarding figure 1, a general overview of EVs’ possible content -regardless of subtype- is illustrated.

Point 2. In the figure 2, the authors show CD14 as a co-receptor of TLR4, whereas in the line 86, the authors use MD-2. The authors may want to keep consistent, at least, using aka such as MD-2 (CD14)?

Response: We wanted to inform you that we have incorporated additional text into the manuscript to clarify the LPS recognition process by TLR4, which describes the initial interaction of LPS with CD14 leading to LPS’s final uptake by the TLR4-MD2 complex, as illustrated in fugure 2. We believe that this addition enhances the clarity of our manuscript and provides a more detailed explanation of the LPS recognition process.

Point 3.  In the LPS tolerance mechanism part, it would be better to have a figure to show the mechanisms as described in the text part.

Response: Once again, we thank you for your valuable input, which will undoubtedly enhance the comprehensiveness and clarity of our manuscript. We have included a figure illustrating the mechanisms implicated in TLR4 tolerance at the level of TLR4 signaling pathway regulation.

Point 4. In the therapeutic potential of EV by targeting TLR4 in HCC or its prevention, the authors may have a short paragraph in introducing the utility of miR-34a in HCC treatment, given TLR4 is a target of TLR4. There are some previous studies showing the effect of miR-34a on tumor proliferation, invasion and metastasis (e.g., 10.1038/s41416-020-0802-1; PMID: 31114344; 10.3389/fcell.2021.640587).

Response: Thank you for your suggestion. We have addressed your recommendation by incorporating a brief paragraph introducing the utility of miR-34a in HCC treatment, emphasizing its impact on tumor proliferation, invasion and metastasis.

Point 5. line 554-556, the authors may make some comments on how TLR4 affect mRNA-based neoantigen vaccines in immunotherapy against HCC.

Response: Thank you for your valuable comment regarding lines 554-556. I've taken your suggestion into consideration and expanded on how TLR4 may influence mRNA-based neoantigen vaccines in the context of HCC immunotherapy. Your input has contributed to a more comprehensive discussion of this topic.

Yours sincerely,

Stamatios Theocharis MD, PhD

Professor of Pathology

First Department of Pathology

Laiko General Hospital

National and Kapodistrian University of Athens

Mikras Asias 75, 11527, Athens, Greece

Tel: +30 2107462002

Email: [email protected], [email protected]

Round 2

Reviewer 1 Report

The manuscript still has some issues, which are still not resolved

1) The title is wordy and does not reflect clearly the subject of the article. I recommend to change the title and make it more relevant to the contents

2) Every chapter should describe the role of vesicles to be removed from the review.

3) The authors should explain in the introduction what new information will get the reader from the article, compared to the other multiple reviews available in scholarly journals. 

Author Response

First Department of Pathology

Laiko General Hospital

National and Kapodistrian University of Athens

September 25th, 2023

Dear Editor,

Response: Interplay of Extracellular Vesicles and TLR4 Signaling in Hepatocellular Carcinoma Pathophysiology and Therapeutics

We thank you and the Reviewers for carefully evaluating our manuscript and for their positive and constructive feedback. Hopefully, our responses below address the points raised by the reviewers. All changes made are presented in the revised manuscript via the “track changes” option in MS word.

Reviewer 1: The manuscript still has some issues, which are still not resolved

Response:

Dear Reviewer,

Thank you for your feedback on our manuscript. We appreciate your input and are committed to improving the quality and clarity of our work.

We have carefully addressed your comments, aiming to make the text more straightforward and to enhance its overall readability.

We take your suggestions seriously and your constructive feedback was invaluable in helping us refine our work in order to deliver a high-quality manuscript.

Point 1: The title is wordy and does not reflect clearly the subject of the article. I recommend to change the title and make it more relevant to the contents

Response:

Dear Reviewer,

Thank you for your thoughtful comment regarding the title of our article. We acknowledge your point, and after careful consideration, we agree that the previous title did not effectively represent the content of the manuscript.

Based on your suggestion, we have revised the title to ensure that it aligns more closely with the manuscripts content and provides a clearer indication of the article's content. Therefore, the title has been modified to: “Interplay of Extracellular Vesicles and TLR4 Signaling in Hepatocellular Carcinoma Pathophysiology and Therapeutics” We believe that the new title encapsulates better the essence of our research.

Point 2:

Dear Reviewer,

Thank you for your valuable feedback. In response to your observation regarding the change in the title, we want to clarify that the revised title has indeed improved the alignment between the title and the content of the manuscript. We believe that each paragraph now directly relates to the central theme presented in the new title. For instance, the paragraph discussing LPS tolerance mediated by TLR4 signaling remains relevant, as it contributes to the understanding of the crucial role of TLR4 signaling in liver physiology, which is highlighted in the new illustration we have created, after another reviewer’s suggestion. Paragraph 3, which describes the role of TLR4 signaling in HCC, is also essential for readers to grasp the pivotal role of TLR4 signaling in HCC development. Therefore, we believe that retaining these sections is vital to provide comprehensive coverage of the topic.

Regarding your recommendation to remove chapters that describe the role of vesicles: We understand your point, but we believe that these chapters play a significant role in providing context and understanding the broader implications of our research. Removing them could potentially result in a loss of valuable insights into the topic.

Point 3:

Dear reviewer,

Thank you for your valuable input. We agree with your suggestion and have addressed it by providing a clear explanation in the introduction section of our article regarding the unique contributions and new insights that readers can gain from our work compared to existing reviews in scholarly journals.

Yours sincerely,

Stamatios Theocharis MD, PhD

Professor of Pathology

First Department of Pathology

Laiko General Hospital

National and Kapodistrian University of Athens

Mikras Asias 75, 11527, Athens, Greece

Tel: +30 2107462002

Email: [email protected], [email protected]

Round 3

Reviewer 1 Report

The authors pretend they do not understand the issues the reviewers point out to them. 

  The reviewer doesn’t see anything new this review could provide to the reader compared to reviews published earlier. This manuscript, in its current form and due to its structure, is not worth publishing in the Pharmaceutics.    Sincerely, 

Author Response

We look forward to hearing from you soon and hope for a favorable decision on the revised manuscript.

Warm regards,